# Heteroscedastic Variational Bayesian Last Layers: Modeling Input-Dependent Noise in Sparse-Data Regression

## Abstract

Bayesian Neural Networks (BNNs) have been extensively studied for uncertainty quantification. To train BNNs efficiently, Variational Bayesian Last Layer (VBLL) provides a sampling-free, deterministic method, significantly reducing computational cost. However, these existing methods assume homoscedastic noise and sufficient data, while real-world industrial applications frequently encounter heteroscedastic noise, where the uncertainty level (i.e., noise) varies with input, and collecting training data in such cases is often expensive. Modeling heteroscedastic noise with sparse data is challenging, but it plays a critical role in setting appropriate safety margins for industrial applications. In this work, we propose Heteroscedastic VBLL (HVBLL) to effectively capture the input-dependent noise. We showcase the impact of noise prior on sparse-data regression, and further design a clustering-based noise level estimation method to provide reliable priors. Experimental results demonstrate that our proposed methods significantly improve the performance of BNNs under heteroscedastic and sparse-data conditions.

## 1 Introduction

Uncertainty quantification is essential for improving the reliability of optimization and decision-making processes, especially in scientific and engineering applications (Smith, 2024). Many models have been studied to accurately characterize uncertainty under the assumption of homoscedasticity (MacKay, 1995; Watson et al., 2021) or in data-rich scenarios (Abdar et al., 2021). However, these assumptions may not hold in real-world settings (Smith, 2024). For example, in aircraft design, the design variables (inputs) are usually geometric parameters. The uncertainty of performance and safety is introduced by stochastic material properties, defects, environments, etc. (Beran et al., 2017; Montomoli et al., 2015). The level of uncertainty usually differs across designs. Some designs may be sensitive to unavoidable sources of uncertainty, which may lead to unsafe products. Therefore, it is essential to quantify the **heteroscedastic uncertainty**, i.e., input-dependent uncertainty, to identify robust designs and allocate appropriate safety factors. Meanwhile, in many industrial design and application processes, **data can be expensive or difficult to obtain**. The sparsity of data brings additional challenges for modeling and uncertainty quantification.

Several classes of methods have been proposed for uncertainty estimation, each with advantages and drawbacks. Monte Carlo Dropout (Gal & Ghahramani, 2016) interprets dropout as approximate Bayesian inference and estimates predictive uncertainty through stochastic forward passes. Gaussian Process (GP) regression (Goldberg et al., 1997; Le et al., 2005) provides a principled Bayesian framework with well-calibrated uncertainty estimates, but its cubic complexity in data size makes it difficult to scale to large datasets. Bayesian Neural Networks (BNNs) (Neal, 2012) place distributions over network weights to capture model uncertainty

(Seitzer et al., 2022; Immer et al., 2023; Deka et al., 2024), but often suffer from high computational overhead, making them difficult to scale to large models and datasets.

However, these approaches often have high computational costs due to reliance on sampling-based inference, or have significant architectural modifications, making them less practical for deployment in large-scale or latency-sensitive applications (Lampinen & Vehtari, 2001; Jospin et al., 2022). The recently proposed VBLL (Harrison et al., 2024) addresses these limitations by offering efficient uncertainty estimation with minimal overhead. This makes VBLL a promising foundation for extending to heteroscedastic problems.

In this work, we propose Heteroscedastic Variational Bayesian Last Layers (HVBLL) to overcome the limitations of the homoscedastic noise assumption, while retaining the key advantages of VBLL. We replace the constant noise term with an input-dependent Gaussian distribution parameterized by an auxiliary neural network. To train this model, we derive a variational formulation with a deterministic lower bound on the marginal likelihood. This approach enables efficient, scalable and sampling-free loss computation. This approach also enables HVBLL to disentangle aleatoric uncertainty from epistemic uncertainty, which improves the interpretability of the model. Since HVBLL retains the structure of Bayesian Last Layers, it remains computationally efficient and can be seamlessly integrated into existing neural network architectures with minimal modification.

To demonstrate the performance of our method, especially in sparse-data regression, we designed a typical toy function featuring heteroscedastic noise to test the models. With the data samples generated by the function, we discovered that VBLLs are sensitive to the noise prior, especially in sparse-data scenarios, while our method are more robust to dataset sizes. We further propose a clustering-based algorithm to estimate the average conditional variance of the data, which can serve as a reliable noise prior for BNNs. We tested this algorithm on a dataset generated by multiple test functions, showing that it can generate noise prior closely matching the ground-truth variance. With this noise prior in practice, we compare our proposed HVBLL with VBLL and other baseline methods on open-source benchmark datasets. The results demonstrate that our approach consistently outperforms the alternatives.

In summary, our contributions in this work can be listed as:

- We extend the original VBLL framework by introducing a heteroscedastic noise term, allowing the model to account for variable noise levels that commonly arise in industrial and other real-world applications.

- We show noise prior is critical to heteroscedastic problems, especially in sparse-data regression problems.

- We propose a clustering-based noise level estimation method to provide reasonable noise priors, and demonstrate the performance of HVBLL.

## 2 PRELIMINARIES

In this work, we study the regression problem of a heteroscedastic system:

$$y = f(\mathbf{x}) + \varepsilon; \ \ \varepsilon \sim N(0, \sigma^2(\mathbf{x})), \tag{1}$$

where $\mathbf{x} \in \mathrm{R}^{n_x}$ is the input vector, $y \in \mathrm{R}$ is the scaler output. $f(\mathbf{x})$ is the **mean function**, and $\sigma(\mathbf{x})$ denotes the **noise level**, i.e., the **aleatoric uncertainty**. Then, the conditional probability of output given the inputs is $p(y|\mathbf{x}) = N(f(\mathbf{x}), \sigma^2(\mathbf{x}))$. The data set is denoted as $\mathcal{D} = \{\mathcal{D}_x, \mathcal{D}_y\} = \{(\mathbf{x}_i, y_i)\}_{i=1}^{N_s}$.

## 2.1 UNCERTAINTIES OF STOCHASTIC MODELS

Uncertainty in stochastic models generally falls into two categories: **aleatoric uncertainty** (inherent noise), which arises from inherent data noise and is irreducible; and **epistemic uncertainty** (model uncertainty), which stems from limited knowledge and can be reduced with more data or better modeling (Abdar et al., 2021). In industrial applications, aleatoric uncertainty often defines the necessary safety margin. Therefore, it is crucial to disentangle aleatoric uncertainty from epistemic uncertainty and to minimize the latter during modeling. The BNN framework offers a principled approach for modeling two uncertainties separately, offering benefits for industrial applications (Jospin et al., 2022).

Given a dataset $\mathcal{D}$, the law of total variance describes the different sources of data variance. It states that if $X, Y$ are the random variables and the variance of $Y$ is finite, then,

$$\text{Var}_{y \sim p(y)}(Y) = \text{E}_{\mathbf{x} \sim p(\mathbf{x})}[\text{Var}_{y \sim p(y|\mathbf{x})}(Y|X)] + \text{Var}_{\mathbf{x} \sim p(\mathbf{x})}[\text{E}_{y \sim p(y|\mathbf{x})}(Y|X)]. \tag{2}$$

Applying Eq. 1 to Eq. 2, it becomes $V_{\text{total}} = E_{\text{noise}} + V_{\text{mean}}$, where $V_{\text{total}}$ is the total variance of $Y$, $E_{\text{noise}} = \text{E}_{\mathbf{x} \sim p(\mathbf{x})}[\sigma^2(\mathbf{x})]$ is the **average aleatoric uncertainty**, $V_{\text{mean}} = \text{Var}_{\mathbf{x} \sim p(\mathbf{x})}[f(\mathbf{x})]$ is the variance of the mean function $f(\mathbf{x})$. $V_{\text{noise}} = \text{Var}_{\mathbf{x} \sim p(\mathbf{x})}[\sigma^2(\mathbf{x})]$ is the variance of aleatoric uncertainty in the input space. Therefore, The magnitude of $V_{\text{noise}}$ and $E_{\text{noise}}$ together characterize the degree of heteroscedasticity in the dataset.

## 2.2 VARIATIONAL BAYESIAN LAST LAYER

For regression tasks described in Eq. 1, canonical Bayesian Last Layers (BLLs) (Brosse et al., 2020; Fiedler & Lucia, 2023) apply a Bayesian treatment to the last layer of neural networks while keeping the feature extractor deterministic. Instead of using fixed weights in the last layer, BLLs place a Gaussian distribution over the weights, resulting in a lightweight and scalable approach. This formulation corresponds to Bayesian linear regression, defined as:

$$y = \mathbf{w}^T \phi_\theta(\mathbf{x}) + \varepsilon; \quad \varepsilon \sim N(0, \sigma^2), \tag{3}$$

where $\phi := \phi_\theta(\mathbf{x}) \in \mathbb{R}^{n_f}$ is referred as *features*. They assume the noise $\varepsilon$ to follow **an i.i.d. Gaussian distribution**, which represents a **homoscedastic model**. Assume $p(\varepsilon) = N(0, \sigma_0^2)$ is the prior noise distribution. A Gaussian prior is placed over the weights, $p(\mathbf{w}) = N(\mu_{w,0}, S_{w,0})$, independently of the noise. Given a Gaussian posterior over the weights, $N(\mu_w, S_w)$, the resulting predictive distribution is $p(y|\mathbf{x}, \theta, \eta) = N(\mu_w^T \phi, \phi^T S_w \phi + \sigma^2)$, where $\eta = (\mu_w, S_w)$ denotes the posterior parameters of the weight distribution. A full training strategy optimizes the last layer variational posterior together with MAP estimation of the features. Its loss function (minimization) is

$$\text{loss}_{\theta,\eta,\sigma^2} = -\mathcal{L}(\theta, \eta, \sigma^2) + N_s^{-1}[-\log p(\theta) - \log p(\sigma^2) + \text{KL}(q_\eta(\mathbf{w})||p(\mathbf{w}))], \tag{4}$$

where the evidence lower bound (ELBO) is

$$\mathcal{L}(\theta, \eta, \sigma^2) = \frac{1}{N_s} \sum_{i=1}^{N_s} \left[ \log N(y_i | \mu_w^T \phi_i, \sigma^2) - \frac{1}{2\sigma^2} \phi_i^T S_w \phi_i \right]. \tag{5}$$

## 3 METHODOLOGY

### 3.1 HETEROSCEDASTIC VARIATIONAL BAYESIAN LAST LAYER

We extend the i.i.d noise term of VBLL described in Eq. 3 to a heteroscedastic noise, which is described in Eq. 1. Then, the Heteroscedastic VBLL (HVBLL) is described as

$$y = \mathbf{w}^T \phi_\theta(\mathbf{x}) + \varepsilon(\mathbf{x}); \ \ \varepsilon \sim N(0, \sigma(\mathbf{x})^2), \tag{6}$$

where noise $\varepsilon(\mathbf{x})$ is assumed to be independent of $\mathbf{w}$. The variance of noise is modeled with a neural network, $\sigma(\mathbf{x})^2 = \exp g_\beta(\mathbf{x})$, where $\beta$ is the weights of the neural network $g$. Then, the predictive distribution (likelihood) of the model is

$$p(y|\mathbf{x}, \theta, \eta, \beta) = N(\mu_w^T \phi, \phi^T S_w \phi + \exp g_\beta(\mathbf{x})). \tag{7}$$

Note that the $\phi^T S_w \phi$ term is the **epistemic uncertainty**, and the $\sigma^2(\mathbf{x})$ is the heteroscedastic **aleatoric uncertainty**. Then, HVBLL employs a sampling-free stochastic variational inference (Hoffman et al., 2013) for the BLL networks. VBLL jointly computes an approximate last layer posterior and optimize network weights by maximizing lower bounds on marginal likelihood. So that the training efficiency is significantly improved, comparing to the Monte-Carlo sampling. Denote the approximate posterior of weights as $q_\eta(\mathbf{w}) = N(\mu_w, S_w)$. Then, the evidence lower bound (ELBO) can be derived from:

$$N_s^{-1} \log p(\mathcal{D}_y | \mathcal{D}_x, \theta, \sigma^2) \geq \mathcal{L}(\theta, \eta, \sigma^2) - N_s^{-1} \mathrm{KL}[q_\eta(\mathbf{w}) || p(\mathbf{w})]. \tag{8}$$

Eq. 8 holds with

$$\mathcal{L}(\theta, \eta, \beta) = \frac{1}{N_s} \sum_{i=1}^{N_s} \left[ \log N(y_i | \mu_w^T \phi_i, g_\beta(\mathbf{x})) - \frac{1}{2 \exp g_\beta(\mathbf{x})} \phi_i^T S_w \phi_i \right], \tag{9}$$

More proof can be found in Appendix A. Then, the loss function is

$$\mathcal{J}_{\theta,\eta,\beta} = -\mathcal{L}(\theta, \eta, \beta) + N_s^{-1}[-\log p(\theta) + \mathrm{KL}(q_\beta(\varepsilon) || p(\varepsilon)) + \mathrm{KL}(q_\eta(\mathbf{w}) || p(\mathbf{w}))], \tag{10}$$

where $p(\varepsilon) = N(0, \sigma_0^2)$ is the prior noise distribution, $q_\beta(\varepsilon) = N(0, \exp g_\beta(\mathbf{x}))$ is the approximated noise distribution. A simple isotropic zero-mean Gaussian priors on feature weights (yielding weight decay regularization). For Gaussian priors, the Kullback–Leibler divergence can be computed in closed form:

$$\mathrm{KL}(q_\eta(\mathbf{w}) || p(\mathbf{w})) = \frac{1}{2} \mathrm{tr}(S_{w,0}^{-1} S_w) + \frac{1}{2} (\mu_w - \mu_{w,0})^T S_{w,0}^{-1} (\mu_w - \mu_{w,0}) - n_x - \log(|S_w|/|S_{w,0}|). \tag{11}$$

### 3.2 CLUSTERING-BASED NOISE LEVEL ESTIMATION

As will be demonstrated in Section 4.2, the performance of heteroscedastic models is strongly affected by the noise prior $\sigma_0^2$, especially when the training data is scarce. Therefore, it is crucial to provide a reasonable estimation of $\sigma_0^2$ given a data set $\mathcal{D}$. According to Eq. 2, the average conditional variance $E_{\mathrm{noise}}$ can be a good estimation of $\sigma_0^2$. We propose Clustering-Based Noise Level Estimation (Algorithm 1) to estimate

$E_{\text{noise}}$, $V_{\text{mean}}$ and $V_{\text{noise}}$ directly from data set $\mathcal{D}$, so that the degree of heteroscedasticity can be assessed and a reasonable $\sigma_0^2$ can be provided.

The proposed algorithm begins by partitioning the dataset into small groups, each containing at most $n_s$ samples. Within each group, samples are selected to have similar inputs $\mathbf{x}$, i.e., they are locally clustered in the input space. Then, we can assume a zero-order approximation of the regression problem in Eq. 1, where the variation of the mean function in each group is negligible, i.e., $V_{\text{mean}} \to 0$. Consequently, the variance of $y$ within each group (denoted as $v_i$) can be interpreted as the local aleatoric uncertainty.

---

**Algorithm 1** Clustering-based noise level estimation

---

**Require:** Data set $\mathcal{D}$
1: **Input:** The maximum number of samples in a group for clustering $n_s$
2: Calculate the total variance $V_{\text{total}}$ of $y$ in $\mathcal{D}$
3: Group samples by similar $\mathbf{x}$ using clustering (e.g., Nearest Neighbors)
4: $N_{\text{group}} \leftarrow$ number of groups
5: **while** $i = 1, ..., N_{\text{group}}$ **do**
6:    $m_i \leftarrow$ mean of $y$ in the cluster group
7:    $v_i \leftarrow$ variance of $y$ in the cluster group
8: **end while**
9: $V_{\text{mean}} \leftarrow \text{Var}(m_i)$
10: $E_{\text{noise}} \leftarrow \text{Mean}(v_i)$
11: $V_{\text{noise}} \leftarrow \text{Var}(v_i)$

---

## 4 EXPERIMENTS

Our experiments aim to demonstrate and validate the following claims: (1) For heteroscedastic regression problems, HVBLL can accurately learn input-dependent noise, whereas VBLL can only capture the average noise; (2) The noise prior is a critical hyperparameter for both HVBLL and VBLL, and their performance on sparse data regressions is highly sensitive to its value — ideally, a noise prior of the same order of magnitude as the ground truth should be provided; (3) We propose a simple algorithm to estimate a reliable noise prior, which performs well under both high-dimensional and sparse-data settings; (4) We compare HVBLL, VBLL, and six baseline models on various real-world datasets, including both sparse and sufficient data scenarios as well as datasets with different degrees of heteroscedasticity, demonstrating the strong and robust performance of HVBLL.

We first use toy functions in Sections 4.1–4.2 where the ground-truth noise distribution is known, allowing for direct validation of model performance. We then evaluate the proposed model and baselines on standard benchmarks for Bayesian neural network regression using datasets from the UCI Machine Learning Repository (Dua & Graff, 2017), European Centre for Medium-Range Weather Forecasts Reanalysis v5 (ERA5) dataset (Hersbach et al., 2020) and a custom dataset on composite structure failure (Appendix J).

### 4.1 HETEROSCEDASTIC TOY FUNCTIONS

Toy functions are used to demonstrate the necessity of employing HVBLL for heteroscedastic problems. The toy function is described as

$$y = f(\mathbf{x}) + g(\mathbf{x})\epsilon; \;\; \epsilon \sim N(0, 1), \mathbf{x} \in [0, 1]^{n_x}. \tag{12}$$

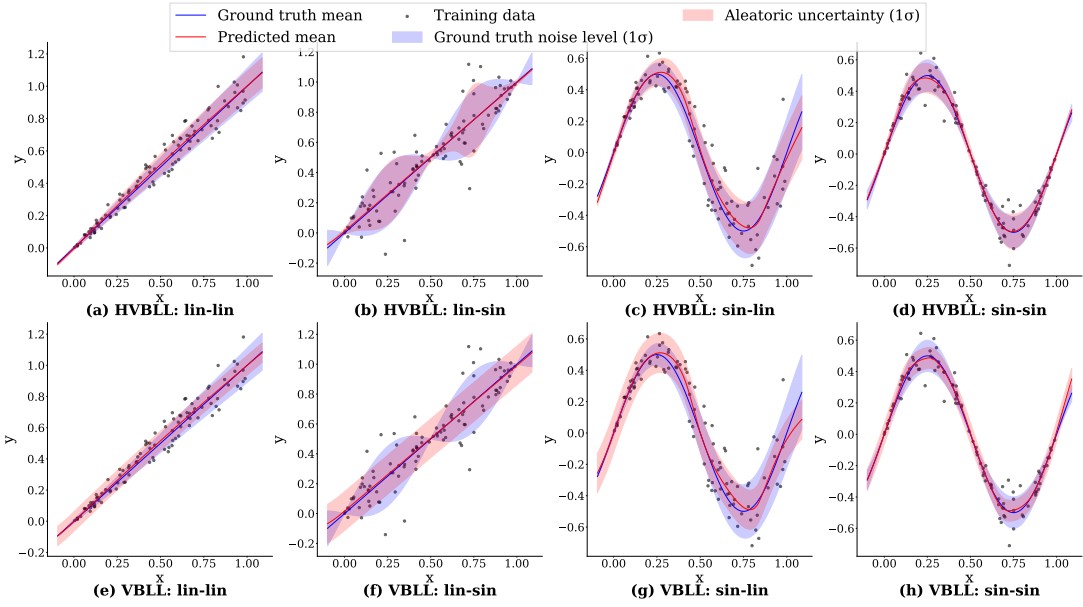

Figure 1: The HVBLL and VBLL are trained on four heteroscedastic toy functions. The blue solid line represents the ground truth mean function $f(x)$, and the blue shaded region indicates the ground truth $1\sigma$ confidence interval, $g(x)$. The red dashed line shows the predicted mean, with the red shaded area representing the predicted aleatoric uncertainty at the $1\sigma$ level.

Then, the conditional probability becomes $p(y|\mathbf{x}) = N(f(\mathbf{x}), g^2(\mathbf{x}))$. The toy functions are combinations of linear functions and sine functions, as shown in Appendix B. In this section, $n_x = 1$, and 200 data points are sampled from a uniform distribution $\mathbf{x} \sim U([0, 1])$ and a Gaussian distribution $\epsilon \sim N(0, 1)$.

Both VBLL and HVBLL are trained on the toy functions. Their dimension of features is $n_f = 32$, the neural network of $\phi_\theta$ contains one hidden layer with 32 neurons. The neural network of $g_\beta$ in HVBLL contains two hidden layers with eight neurons. The noise prior $\sigma_0^2 = 0.01$. The initial learning rate is $0.01$, the learning rate gradually reduces during the training of 5,000 epochs. Adam optimizer (Kingma & Ba, 2014) is used for training.

Their performance is illustrated in Fig. 1. The results demonstrate that HVBLL effectively captures the heteroscedastic noise in the toy functions, whereas the VBLL can only estimate the average noise level. For example, by comparing (a) and (e), we observe that VBLL fails to capture the characteristic of smaller variance when $x$ is small, whereas HVBLL successfully captures this heteroscedastic behavior. Comparing (b) and (f), the variance of our designed toy function varies with input $x$ in a cosine-like pattern. HVBLL accurately fits this wave-like variance, while VBLL still treats it as homoscedastic, and thus cannot distinguish the variance differences.

## 4.2 SPARSE DATA SCENARIOS

A more complex toy function (Eq. 13) is used to study the influence of noise prior $\sigma_0^2$ on VBLL and HVBLL, especially in the sparse data scenarios. In this section, 20 and 200 data points are sampled from a uniform distribution $\mathbf{x} \sim U([-0.5, 1.5])$ and a Gaussian distribution $\epsilon \sim N(0, 1)$ for training, 20 other samples are sampled for testing. Both VBLL and HVBLL have their dimension of features $n_f = 16$, the neural network

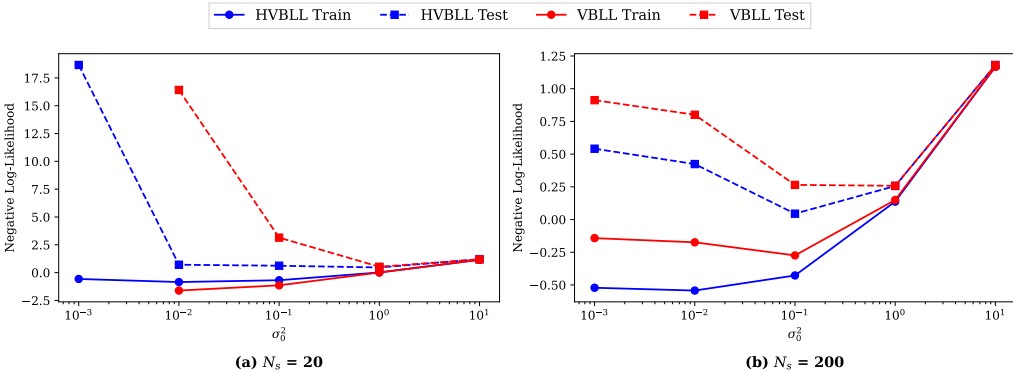

Figure 2: Influence of noise prior $\sigma_0^2$ on HVBLL and VBLL. The left plot shows results under the sparse data setting, while the right plot corresponds to the data-rich setting. Outlier data points with excessively large values are omitted for clarity.

of $\phi_\theta$ contains three hidden layers with 64 neurons. The neural network of $g_\beta$ in HVBLL contains one hidden layer with eight neurons. The initial learning rate is 0.01, the learning rate gradually reduces during the training of 20,000 epochs. Adam optimizer is used for training.

$$
\begin{cases}
f(x) = x^2 \sin(4\pi x) \\
g(x) = 0.05 \max(1.0, 5x + 1)
\end{cases} , \quad x \in [-0.5, 1.5]
\tag{13}
$$

Fig. 2 shows the negative log likelihood on the training and testing sets of VBLL and HVBLL under different values of noise prior ($\sigma_0^2$s). The ground truth $\sigma_{0,\text{true}}^2 = 0.052$. In Fig. 2 (a), the values of 'HVBLL Test', 'VBLL Train' and 'VBLL Test' in $\sigma_0^2 = 10^{-3}$ are omitted because they are several orders of magnitude larger than the remaining cases, i.e., the models overfit the problem in these cases. The results indicate that a small $\sigma_0^2$ leads to overfitting, because the model assumes the data is nearly noise-free. In contrast, a large $\sigma_0^2$ leads to underfitting of the noise level. The performance of VBLL and HVBLL in sparse data scenarios ($N_s = 20$) is shown in Fig. 4 (Appendix C). This sensitivity to $\sigma_0^2$ is significantly reduced when more data is available, as shown in Fig. 5 (Appendix C), where $N_s = 200$.

In summary, both VBLL and HVBLL are sensitive to the noise prior in sparse data scenarios. A noise prior that is too small can lead to severe overfitting; however, when it is on the same order of magnitude as the true noise level—or one or two orders of magnitude larger—it can yield better fitting performance.

### 4.3 ESTIMATION OF NOISE LEVEL

As demonstrated in Section 4.2, providing an appropriate noise prior $\sigma_0^2$ is crucial for both VBLL and HVBLL. Based on Eq.2, $E_{\text{noise}}$ is a suitable choice for $\sigma_0^2$, so we introduce Algorithm1 to estimate $E_{\text{noise}}$. This algorithm is evaluated on the four multivariate toy functions described in Section 4.1, under varying input dimensions $n_x$ and sample sizes $N_s$. For comparison, we also employ Monte Carlo Dropout (MC-Dropout, (Gal & Ghahramani, 2016)), a widely-used deep learning method for uncertainty estimation, to estimate the noise level and benchmark its performance against our proposed approach.

The ratios of estimated $E_{\text{noise}}$ to the real value of all cases are plotted in Fig. 3. The details are presented in Table. 3 (Appendix D). The results indicate that our algorithm estimates $E_{\text{noise}}$ to be within the same order of magnitude as the true value, even under sparse data conditions and in high-dimensional settings. In contrast,

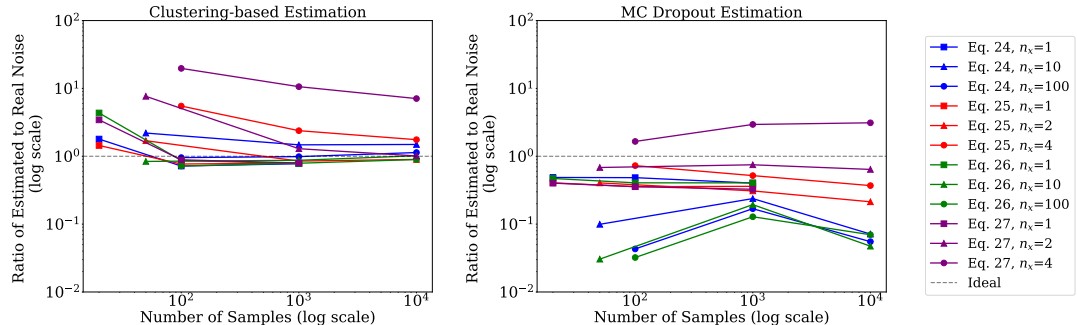

Figure 3: Estimation of noise. Algorithm 1 (left) and MC-Dropout (right) are tested on four toy functions (Eq. 28 - 31) with different input dimensions and sample sizes. The plots show the ratio of the estimated $E_{\text{noise}}$ to the real value, where a ratio of one (gray dashed line) indicates a perfect estimation. Curves closer to the dashed line represent more accurate noise estimates.

MC-Dropout exhibits larger estimation errors. Notably, our method tends to slightly overestimate the noise level, whereas MC-Dropout underestimates it. Based on the results in Section 4.2, overestimation of noise prior generally yields better fitting performance. Therefore, Algorithm 1 is valid to estimate $\sigma_0^2$.

### 4.4 REAL-WORLD DATASETS

We evaluate HVBLL against VBLL and six baselines on a wide range of benchmarks. The six baseline models are: Monte-Carlo Dropout (Dropout, (Gal & Ghahramani, 2016)), Stochastic Weight Averaging Gaussian (SWAG, (Maddox et al., 2019)), Latent Derivative Bayesian Last Layer Networks (BLL, (Watson et al., 2021)), Heteroscedastic Uncertainty Estimation with Probabilistic Neural Networks (PNN, (Seitzer et al., 2022)), Deterministic Variational Inference (DVI, (Wu et al., 2018)) and Mixture Density Networks (MDN, (Bishop, 1994)).

The benchmarks cover four UCI datasets and their modified variants (Appendix E), the ERA5 dataset (Appendix F), and a custom dataset on composite structure failure (Appendix G). To examine robustness under varying data availability, each case is tested with three different training sample sizes ($N_s$). Performance is evaluated using Negative Log Likelihood (NLL), Mean Absolute Error (MAE), and Continuous Ranked Probability Score (CRPS), all of which are preferable when lower. Each experiment is repeated ten times with different random samplings.

Tables 1 and 2 summarize the advantage of HVBLL over the baselines. Table 1 focuses on sparse-data scenarios, while Table 2 reports the average advantage across all scenarios.

For each metric, HVBLL's improvement over each baseline is computed on a case-by-case basis. NLL improvements are reported as absolute differences due to scale invariance, whereas MAE and CRPS improvements are expressed as relative percentages since they are scale-dependent. The Average Improvement (AI) represents the relative improvement of HVBLL over each baseline model (positive values indicate HVBLL performs better). The Win Rate (WR) shows the percentage of cases where HVBLL outperforms the baseline model. All metrics are calculated on test data across dataset cases. Further details of network architectures, training setups and results are provided in Appendix H–J.

Overall, HVBLL outperforms VBLL and other baselines in most cases, with particularly strong improvements on sparse-data cases. The advantage is especially pronounced under stronger heteroscedastic noise and limited

training data. HVBLL demonstrates robustness across dataset sizes and input dimensionalities, maintaining reliable predictions even in sparse-data settings.

Table 1: HVBLL Advantage Summary (small datasets)

| Model | NLL | | MAE | | CRPS | | $N_{\text{case}}$ |
|---|---|---|---|---|---|---|---|
| | AI | WR | AI | WR | AI | WR | |
| VBLL | 0.553 | 81.0 % | 0.073 | 66.7 % | 0.113 | 66.7 % | 21 |
| BLL | 9.367 | 100.0 % | 0.365 | 100.0 % | 0.548 | 100.0 % | 21 |
| MC-Dropout | 1.777 | 90.5 % | 0.020 | 61.9 % | 0.152 | 81.0 % | 21 |
| PNN | 11.261 | 100.0 % | 0.041 | 71.4 % | 0.080 | 85.7 % | 21 |
| SWAG | 2.133 | 100.0 % | 0.380 | 76.2 % | 0.424 | 100.0 % | 21 |
| DVI | 1.311 | 100.0 % | 0.153 | 95.2 % | 0.154 | 95.2 % | 21 |
| MDN | 3.063 | 100.0 % | 0.332 | 90.5 % | 0.533 | 100.0 % | 21 |
| **Overall** | **0.440** | **93.9 %** | **0.195** | **80.3 %** | **0.286** | **89.8 %** | **21** |

Table 2: HVBLL Advantage Summary (all datasets)

| Model | NLL | | MAE | | CRPS | | $N_{\text{case}}$ |
|---|---|---|---|---|---|---|---|
| | AI | WR | AI | WR | AI | WR | |
| VBLL | 0.401 | 79.4 % | 0.023 | 55.6 % | 0.067 | 68.3 % | 63 |
| BLL | 7.457 | 100.0 % | 0.344 | 100.0 % | 0.528 | 100.0 % | 63 |
| Dropout | 2.026 | 88.9 % | -0.018 | 46.0 % | 0.115 | 71.4 % | 63 |
| PNN | 6.234 | 100.0 % | -0.002 | 50.8 % | 0.032 | 65.1 % | 63 |
| SWAG | 1.966 | 98.4 % | 0.374 | 79.4 % | 0.417 | 96.8 % | 63 |
| DVI | 0.635 | 90.5 % | 0.124 | 84.1 % | 0.116 | 81.0 % | 63 |
| MDN | 2.184 | 96.8 % | 0.273 | 77.8 % | 0.466 | 82.5 % | 63 |
| **Overall** | **0.331** | **90.0 %** | **0.160** | **70.5 %** | **0.249** | **80.7 %** | **63** |

## 5  DISCUSSION AND CONCLUSIONS

We introduced the Heteroscedastic Variational Bayesian Last Layer (HVBLL) model to capture input-dependent noise in heteroscedastic regression tasks with high computational efficiency. HVBLL can be easily integrated into existing neural network architectures, making it a practical solution for uncertainty modeling. We evaluated HVBLL on both toy regression functions and benchmark datasets, comparing it with the vanilla VBLL and six other baselines under varying training data sizes. Results demonstrate that the choice of noise prior significantly impacts the performance. To address this, we proposed a clustering-based noise estimation method for more informed prior selection. Overall, HVBLL mostly achieves superior performance, particularly in sparse data scenarios, highlighting its potential for robust uncertainty modeling.

While our method makes a meaningful step forward, it still relies on the assumption of independent of noise and weights. The performance of HVBLL is also affected by the noise prior as shown in our experiments. We proposed a clustering-based strategy to estimate this prior, but the performance of HVBLL on more complex datasets remains an open question. Exploring better noise prior modeling is an important direction for future work.

## ETHICS STATEMENT

This work adheres to the ICLR Code of Ethics. Most experiments are conducted on publicly available benchmark datasets (e.g., UCI repository, ERA5 reanalysis dataset) and synthetically generated toy functions. The custom dataset on composite structure failure will be made available in the supplementary material to facilitate replication and extension of our results. No human subjects, personal data, or sensitive social attributes are involved. The research objective is to advance uncertainty modeling methods for scientific and engineering applications, particularly under sparse-data and heteroscedastic conditions, without potential misuse or discriminatory applications. All datasets are used in accordance with their respective licenses, and no conflicts of interest or undisclosed sponsorships are associated with this work.

## REPRODUCIBILITY STATEMENT

To ensure reproducibility, we provide full details of the model formulation, derivations, and experimental settings in the main text and appendices. The theoretical derivations, including the proof of the evidence lower bound, are presented in Appendix A. The toy functions used for validation are defined in Appendix B, and the benchmark datasets (UCI, ERA5, and composite structure failure dataset) with preprocessing details are described in Appendices E-G. The clustering-based noise prior estimation algorithm is fully specified in Section 3.2 and Appendix D. Network architectures, training hyperparameters, and optimization setups are listed in Appendices C, H-J. Anonymous implementation code and experimental scripts will be made available in the supplementary material to facilitate replication and extension of our results.

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

## A    Evidence Lower Bound of HVBLL

The proof of evidence lower bound of Heteroscedastic Variational Bayesian Last Layers is presented here. Following the same procedure in (Harrison et al., 2024), we consider the more general form of multivariate regression case:

$$\mathbf{y} = W\phi + \varepsilon, \tag{14}$$

where $\varepsilon \sim MN(0, \Sigma)$ is assumed to be independent from $W$, $\theta$ is the parameters of the feature neural network $\phi := \phi_\theta(\mathbf{x})$. Given a parameter $W$, the distribution of $\mathbf{y}$ is

$$p(\mathbf{y}|\mathbf{x}, \theta, W, \Sigma) = N(W\phi, \Sigma). \tag{15}$$

A matrix normal prior is placed on the weights $W \sim MN(\underline{\overline{w}}, I, \underline{S})$. Then, the posterior is also matrix normal. For the details about matrix normal distributions, please refer to (Box & Tiao, 2011). Then, given a parameter distribution $MN(\bar{W}, I, S)$, denote $\eta = \{\bar{W}, S\}$, the predictive distribution (likelihood) of the HVBLL is

$$p(\mathbf{y}|\mathbf{x}, \theta, \eta, \Sigma) = N(\bar{W}\phi, \phi^T S\phi I + \Sigma). \tag{16}$$

Looking at the inequality in Eq. 8 in multivariate regression cases:

$$N_{\mathrm{s}}^{-1} \log p(\mathcal{D}_y|\mathcal{D}_x, \theta, \Sigma) \geq \mathcal{L}(\theta, \eta, \Sigma) - N_{\mathrm{s}}^{-1}\mathrm{KL}[q_\eta(W)||p(W)], \tag{17}$$

we need to prove Eq. 17 holds with the following ELBO:

$$\mathcal{L}(\theta, \eta, \Sigma) = \frac{1}{N_{\mathrm{s}}} \sum_{i=1}^{N_{\mathrm{s}}} \left[ \log N(y_i|\bar{W}\phi_i, \Sigma) - \frac{1}{2}\phi_i^T S\phi_i \mathrm{tr}(\Sigma^{-1}) \right]. \tag{18}$$

*Proof.*

$$\log p(\mathcal{D}_y|\mathcal{D}_x, \theta, \Sigma) = \log \mathrm{E}_{p(W)}\left[ p(\mathcal{D}_y|\mathcal{D}_x, \theta, W, \Sigma) \right] \tag{19}$$

$$= \log \mathrm{E}_{q_\eta(W)}\left[ p(\mathcal{D}_y|\mathcal{D}_x, \theta, W, \Sigma)\frac{p(W)}{q_\eta(W)} \right] \tag{20}$$

$$\geq \mathrm{E}_{q_\eta(W)}\left[ p(\mathcal{D}_y|\mathcal{D}_x, \theta, W, \Sigma) - \mathrm{KL}\left( q_\eta(W)|p(W) \right) \right] \tag{21}$$

$$= \sum_{i=1}^{N_{\mathrm{s}}} \mathrm{E}_{q_\eta(W)}\left[ p(\mathbf{y}_i|\mathbf{x}_i, \theta, W, \Sigma) \right] - \mathrm{KL}\left( q_\eta(W)|p(W) \right). \tag{22}$$

Since the expectation term in Eq. 22 is the log of a Normal distribution, by applying *Lemma 1*, we have

$$\mathrm{E}_{q(W|\eta)}\left[ p(\mathbf{y}_i|\mathbf{x}_i, \theta, W, \Sigma) \right] = \log p(\mathbf{y}_i|\mathbf{x}_i, \theta, W, \Sigma) - \frac{1}{2}\phi_i^T S\phi_i \Sigma^{-1}, \tag{23}$$

which completes the proof. Modeling $\Sigma$ with a neural network using $\mathbf{x}$ as input does not corrupt the proof, therefore, Eq. 9 holds.

**Lemma 1**. *Let* $q(\mu) = N(\bar{\mu}, S)$ *and* $p(y|X, \mu) = N(X\mu, \Sigma)$ *with* $\mathbf{y} \in \mathrm{R}^N$, $\bar{\mu}, \mu \in \mathrm{R}^M$, $X \in \mathrm{R}^{N \times M}$, *and* $S, \Sigma \in \mathrm{R}^{M \times M}$. *Then*

$$\mathrm{E}_{q(\mu)}\left[p(\mathbf{y}|X, \mu)\right] = \log p(\mathbf{y}|X, \bar{\mu}) - \frac{1}{2}\Sigma^{-1}X^T S X, \tag{24}$$

*Proof.* This was proved in *Lemma 4* in (Harrison et al., 2024), we repeat their proof here.

$$\mathrm{E}_{q(\mu)}\left[p(\mathbf{y}|X, \mu)\right] = -\frac{1}{2}\mathrm{E}_{q(\mu)}\left[\mathrm{logdet}(2\pi\Sigma) + (\mathbf{y} - X\mu)^T\Sigma^{-1}(\mathbf{y} - X\mu)\right] \tag{25}$$

$$= -\frac{1}{2}\left(\mathrm{logdet}(2\pi\Sigma) + \mathrm{E}_{q(\mu)}\left[(\mathbf{y} - X\mu)^T\Sigma^{-1}(\mathbf{y} - X\mu)\right]\right) \tag{26}$$

$$= -\frac{1}{2}\left(\mathrm{logdet}(2\pi\Sigma) + (\mathbf{y} - X\mu)^T\Sigma^{-1}(\mathbf{y} - X\mu) + \mathrm{tr}(\Sigma^{-1}X^T S X)\right). \tag{27}$$

## B  TOY FUNCTIONS

Four toy functions are used to represent heteroscedastic problems described in Eq. 12. The four toy functions are combinations of linear functions and sine functions, as shown below. The $a = 0.1, 0.2, 0.2, 0.1$ in Eq. 28 - 31, respectively.

$$\begin{cases} f(\mathbf{x}) = \frac{1}{n_x}\sum_{i=1}^{n_x} x_i \\ g(\mathbf{x}) = \frac{a}{n_x}\sum_{i=1}^{n_x} x_i \end{cases} \tag{28}$$

$$\begin{cases} f(\mathbf{x}) = \frac{1}{n_x}\sum_{i=1}^{n_x} x_i \\ g(\mathbf{x}) = a\prod_{i=1}^{n_x} \sin(2\pi x_i) \end{cases} \tag{29}$$

$$\begin{cases} f(\mathbf{x}) = \frac{1}{2}\prod_{i=1}^{n_x} \sin(2\pi x_i) \\ g(\mathbf{x}) = \frac{a}{n_x}\sum_{i=1}^{n_x} x_i \end{cases} \tag{30}$$

$$\begin{cases} f(\mathbf{x}) = \frac{1}{2}\prod_{i=1}^{n_x} \sin(2\pi x_i) \\ g(\mathbf{x}) = a\prod_{i=1}^{n_x} \sin(2\pi x_i) \end{cases} \tag{31}$$

## C  INFLUENCE OF $\sigma_0^2$ IN SPARSE AND SUFFICIENT DATA SCENARIOS

The performance of VBLL and HVBLL in sparse data ($N_\mathrm{s} = 20$) and sufficient data ($N_\mathrm{s} = 200$) scenarios is shown in Fig. 4 and Fig. 5. Comparing to Fig. 4. The results indicates that the sensitivity of performance to the noise prior $\sigma_0^2$ is significantly reduced when more data is available.

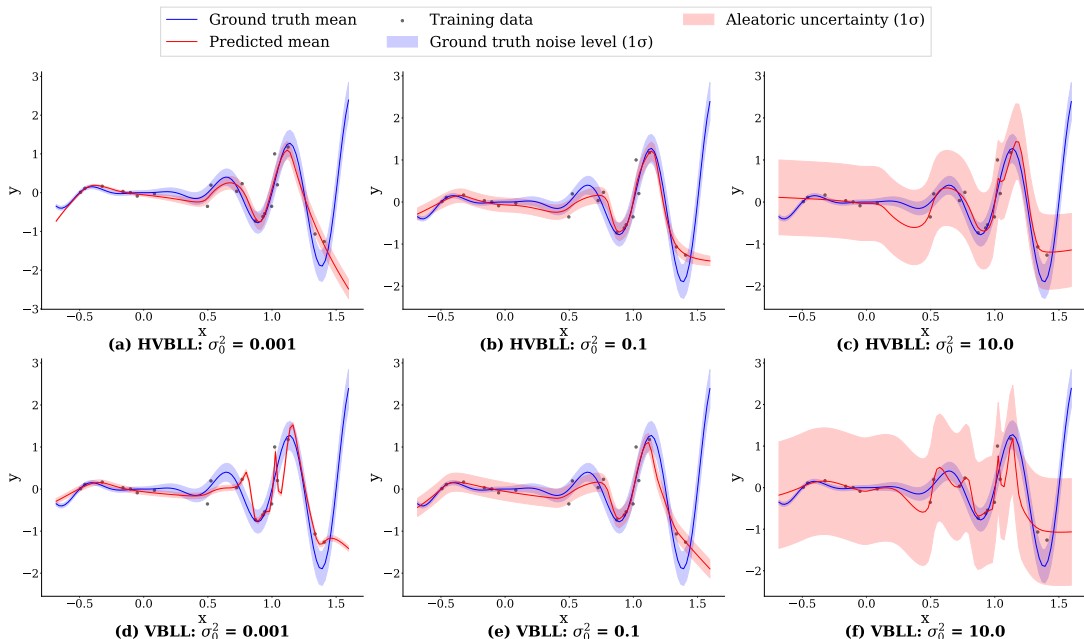

Figure 4: Influence of noise prior $\sigma_0^2$ in sparse data scenarios. The first row shows the results of HVBLL, the second shows VBLL. The three columns show the performance of models trained with different $\sigma_0^2$ values. The solid lines represent the mean function, and the shaded regions indicate the $1\sigma$ confidence interval.

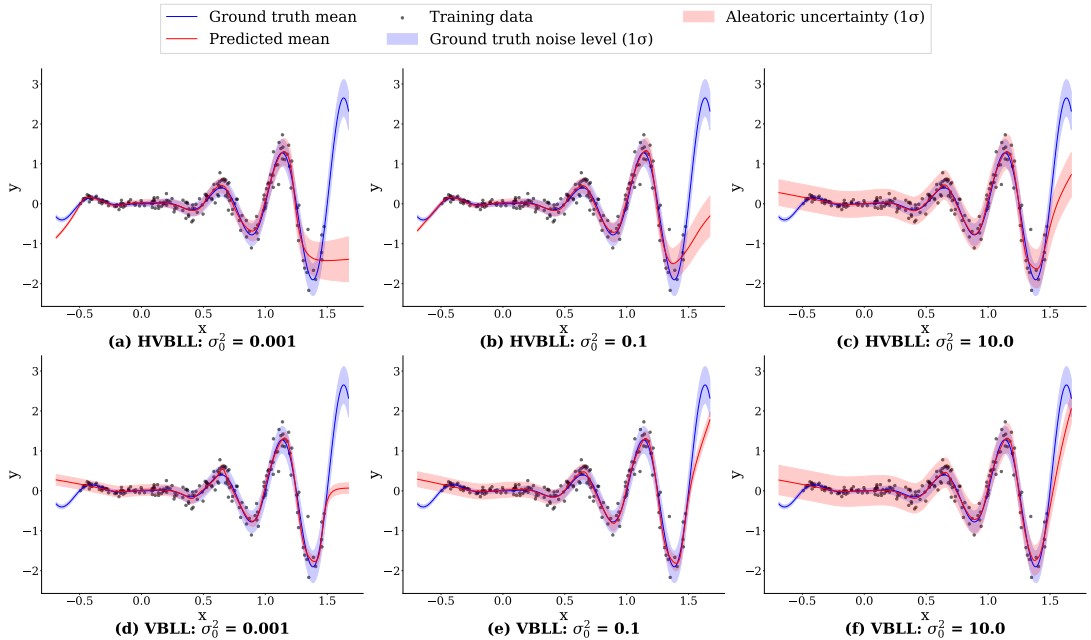

Figure 5: Influence of noise prior $\sigma_0^2$ on VBLL in sufficient data scenarios

## D  ESTIMATION OF VARIANCE IN TOY FUNCTIONS

Algorithm 1 using Nearest Neighbor (denoted by "Our (NN)") and MC-Dropout (denoted by "MC") are used to estimate the noise level in the four toy functions shown in Section 4.1, where $a = 0.1$. Another version of Algorithm 1 using K-means as the clustering technique (denoted by "Our (KM)") is also tested. Different input dimension $n_x$, number of samples $N_s$ are tested on all toy functions. Each case is repeatedly tested on ten randomly sampled data sets. The average values are presented in Table. 3. The maximum number of samples in a group for clustering ($n_s$) is five in Algorithm 1. There is five hidden layers in the neural network of MC-Dropout, with 128 neurons in each layer; the dimension of features is 64. The initial learning rate is 0.01, the learning rate gradually reduces during the training of 20,000 epochs. Adam optimizer is used for training.

Both using Nearest Neighbor and K-means as the clustering technique in Algorithm 1 achieve similar results. Algorithm 1 not only estimates the average noise level $E_{\text{noise}}$, but also provides estimates for the variance of the mean, $V_{\text{mean}}$, and the variance of the noise level, $V_{\text{noise}}$. By comparing the magnitudes of $V_{\text{mean}}$ and $V_{\text{noise}}$, we can preliminarily assess the degree of heteroscedasticity in the problem, and thus determine whether using HVBLL would offer advantages over VBLL.

## E  DESCRIPTION OF MODIFIED UCI DATASETS

Four UCI datasets are used for testing. Table 4 presents the ID, name, input dimension ($n_x$), and number of samples ($N_{\text{s, total}}$) for each dataset. All datasets have output dimension $n_y = 1$. Since the original datasets lack significant heteroscedastic noise, they are not well-suited for highlighting the differences between HVBLL and VBLL. Inspired by **partially observable systems** that frequently occurs in industrial applications, the datasets are modified by treating a subset of the inputs as unobservable inputs (states). These unobservable inputs are assumed to be random variables, and thus become the source of aleatoric uncertainty in the system (Smith, 2024). The $E_{\text{noise}}$, $V_{\text{mean}}$ and $V_{\text{noise}}$ of the original and modified datasets are calculated with Algorithm 1. Table 5 shows the indexes of inputs that are considered as unobservable and the source of uncertainty.

## F  DESCRIPTION OF ERA5 DATASETS

The ERA5 dataset is a global atmospheric reanalysis product providing hourly data on single levels from 1940 to the present, across $\sim 137$ atmospheric levels, with $\sim 31$ km horizontal resolution. In our work, we extract only the 2 m temperature variable over a sub-region bounded by latitudes $7°$ to $83°$ and longitudes $-169°$ to $-35°$, for the years 2020 through 2024. Within each month we sample on days 1, 5, 10, 15, 20, 25, and at times 00:00, 06:00, 12:00, and 18:00 UTC. The raw data therefore forms a spatial-temporal field with shape $(n_{\text{month}}, n_{\text{lat}}, n_{\text{lon}})$, which we resize to $(n_{\text{month}}, 64, 64)$ for the dataset.

In our heteroscedastic Bayesian regression setup, we use month, latitude, and longitude as observed inputs and predict the 2 m temperature. Day and hour are treated as latent (unobserved) variables, whose absence induces input-dependent noise (heteroscedasticity). Accordingly, the model learns not only the conditional mean temperature for each month, latitude, and longitude, but also a non-constant variance of temperature in each month.

Table 3: Variance estimation of multivariate toy functions

| Case | Toy function | $n_x$ | $N_s$ | Real $E_{noise}$ | Our (NN) $E_{noise}$ | $V_{total}$ | $V_{mean}$ | $V_{noise}$ | Our (KM) $E_{noise}$ | MC $E_{noise}$ |
|---|---|---|---|---|---|---|---|---|---|---|
| 1 | Eq. 28 | 1 | 20 | 4.4e-3 | 7.9e-3 | 7.6e-2 | 5.7e-2 | 1.2e-3 | 7.3e-3 | 2.2e-3 |
| 2 | Eq. 28 | 1 | 100 | 4.4e-3 | 3.1e-3 | 8.1e-2 | 7.7e-2 | 7.3e-4 | 2.9e-3 | 2.1e-3 |
| 3 | Eq. 28 | 1 | 1000 | 4.4e-3 | 3.4e-3 | 8.6e-2 | 8.3e-2 | 9.2e-4 | 3.3e-3 | 1.8e-3 |
| 4 | Eq. 28 | 10 | 50 | 2.6e-3 | 5.8e-3 | 1.1e-2 | 4.2e-3 | 6.4e-4 | 5.5e-3 | 2.7e-4 |
| 5 | Eq. 28 | 10 | 1000 | 2.6e-3 | 3.9e-3 | 1.0e-2 | 5.9e-3 | 4.9e-4 | 3.8e-3 | 6.4e-3 |
| 6 | Eq. 28 | 10 | 10000 | 2.6e-3 | 4.0e-3 | 1.0e-2 | 6.2e-3 | 2.4e-4 | 3.9e-3 | 1.9e-4 |
| 7 | Eq. 28 | 100 | 100 | 2.5e-3 | 2.4e-3 | 3.3e-3 | 7.0e-4 | 2.8e-4 | 2.0e-3 | 1.1e-4 |
| 8 | Eq. 28 | 100 | 1000 | 2.5e-3 | 2.4e-3 | 3.3e-3 | 7.0e-4 | 2.9e-4 | 2.0e-3 | 4.3e-4 |
| 9 | Eq. 28 | 100 | 10000 | 2.5e-3 | 2.8e-3 | 3.3e-3 | 4.3e-4 | 1.5e-4 | 2.7e-3 | 1.4e-4 |
| 10 | Eq. 29 | 1 | 20 | 4.9e-3 | 7.2e-3 | 8.1e-2 | 6.3e-2 | 1.2e-3 | 6.3e-3 | 2.0e-3 |
| 11 | Eq. 29 | 1 | 100 | 4.9e-3 | 3.8e-3 | 8.0e-2 | 7.7e-2 | 1.0e-3 | 3.7e-3 | 1.8e-3 |
| 12 | Eq. 29 | 1 | 1000 | 4.9e-3 | 3.9e-3 | 8.6e-2 | 8.2e-2 | 1.1e-3 | 3.8e-3 | 1.8e-3 |
| 13 | Eq. 29 | 2 | 50 | 2.4e-3 | 4.2e-3 | 3.9e-2 | 3.1e-2 | 6.2e-4 | 4.2e-3 | 1.0e-3 |
| 14 | Eq. 29 | 2 | 1000 | 2.4e-3 | 2.1e-3 | 4.4e-2 | 4.1e-2 | 7.2e-4 | 2.0e-3 | 7.8e-4 |
| 15 | Eq. 29 | 2 | 10000 | 2.4e-3 | 2.2e-3 | 4.4e-2 | 4.1e-2 | 7.7e-4 | 2.2e-3 | 5.3e-4 |
| 16 | Eq. 29 | 4 | 100 | 6.2e-4 | 3.4e-3 | 2.2e-2 | 1.8e-2 | 4.1e-4 | 3.4e-3 | 4.5e-4 |
| 17 | Eq. 29 | 4 | 1000 | 6.2e-4 | 1.4e-3 | 2.1e-2 | 1.9e-2 | 2.4e-4 | 1.4e-3 | 3.3e-4 |
| 18 | Eq. 29 | 4 | 10000 | 6.2e-4 | 1.0e-3 | 2.1e-2 | 2.0e-2 | 1.7e-4 | 1.0e-3 | 2.3e-4 |
| 19 | Eq. 30 | 1 | 20 | 4.4e-3 | 1.9e-2 | 1.1e-1 | 9.2e-2 | 2.9e-3 | 2.0e-2 | 2.1e-3 |
| 20 | Eq. 30 | 1 | 100 | 4.4e-3 | 3.8e-3 | 1.2e-1 | 1.1e-1 | 7.7e-4 | 2.2e-3 | 1.8e-3 |
| 21 | Eq. 30 | 1 | 1000 | 4.4e-3 | 3.4e-3 | 1.2e-1 | 1.2e-1 | 9.2e-4 | 3.3e-3 | 1.8e-3 |
| 22 | Eq. 30 | 10 | 50 | 2.6e-3 | 2.2e-3 | 2.7e-3 | 4.7e-4 | 2.7e-4 | 2.0e-3 | 8.2e-5 |
| 23 | Eq. 30 | 10 | 1000 | 2.6e-3 | 2.3e-3 | 2.8e-3 | 5.8e-4 | 3.2e-4 | 2.2e-3 | 5.2e-4 |
| 24 | Eq. 30 | 10 | 10000 | 2.6e-3 | 2.7e-3 | 2.9e-3 | 3.0e-4 | 2.2e-4 | 2.6e-3 | 1.3e-4 |
| 25 | Eq. 30 | 100 | 100 | 2.5e-3 | 1.8e-3 | 2.3e-3 | 4.2e-4 | 2.1e-4 | 1.5e-3 | 8.1e-5 |
| 26 | Eq. 30 | 100 | 1000 | 2.5e-3 | 1.9e-3 | 2.4e-3 | 4.7e-4 | 2.3e-4 | 1.6e-3 | 3.2e-4 |
| 27 | Eq. 30 | 100 | 10000 | 2.5e-3 | 2.2e-3 | 2.5e-3 | 2.5e-4 | 1.2e-4 | 2.1e-3 | 1.8e-4 |
| 28 | Eq. 31 | 1 | 20 | 4.9e-3 | 1.7e-2 | 1.0e-1 | 8.4e-2 | 2.3e-3 | 1.8e-2 | 2.0e-3 |
| 29 | Eq. 31 | 1 | 100 | 4.9e-3 | 4.4e-3 | 1.2e-1 | 1.2e-1 | 8.5e-4 | 4.1e-3 | 1.8e-3 |
| 30 | Eq. 31 | 1 | 1000 | 4.9e-3 | 3.9e-3 | 1.2e-1 | 1.2e-1 | 1.1e-3 | 3.8e-3 | 1.6e-3 |
| 31 | Eq. 31 | 2 | 50 | 2.4e-3 | 1.9e-2 | 5.7e-2 | 4.1e-2 | 2.4e-3 | 1.9e-2 | 1.7e-3 |
| 32 | Eq. 31 | 2 | 1000 | 2.4e-3 | 3.2e-3 | 6.4e-2 | 6.1e-2 | 7.1e-4 | 3.0e-3 | 1.9e-3 |
| 33 | Eq. 31 | 2 | 10000 | 2.4e-3 | 2.5e-3 | 6.4e-2 | 6.2e-2 | 6.7e-4 | 2.5e-3 | 1.6e-3 |
| 34 | Eq. 31 | 4 | 100 | 6.2e-4 | 1.2e-2 | 1.5e-2 | 6.3e-3 | 2.1e-3 | 9.5e-3 | 1.0e-3 |
| 35 | Eq. 31 | 4 | 1000 | 6.2e-4 | 6.6e-3 | 1.6e-2 | 1.1e-2 | 1.6e-3 | 5.5e-3 | 1.8e-3 |
| 36 | Eq. 31 | 4 | 10000 | 6.2e-4 | 4.4e-3 | 1.6e-2 | 1.2e-2 | 9.6e-4 | 4.1e-3 | 1.9e-3 |

Table 4: UCI datasets for testing

| ID | Name | $n_x$ | $N_s$ |
|---|---|---|---|
| 165 | Concrete Compressive Strength (Yeh, 1998) | 8 | 1030 |
| 186 | Wine Quality (Cortez & Reis, 2009) | 11 | 4898 |
| 291 | Airfoil Self-Noise (Brooks & Marcolini, 1989) | 5 | 1503 |
| 294 | Combined Cycle Power Plant (Tfekci & Kaya, 2014) | 4 | 9568 |

Table 5: Details and variance estimation of testing datasets

| Dataset ID | UCI ID | $n_x$ | $E_{\text{noise}}$ | $V_{\text{mean}}$ | $V_{\text{noise}}$ | Indexes of omitted inputs | $N_s$ of three datasets |
|---|---|---|---|---|---|---|---|
| 1 | 165 | 8 | 6.2e+1 | 2.1e+2 | 1.4e+1 | N/A | 200, 500, 1000 |
| 2 | 165 | 6 | 7.7e+1 | 1.9e+2 | 2.0e+1 | (0,3) | " |
| 3 | 165 | 5 | 8.6e+1 | 1.8e+2 | 2.3e+1 | (0,3,6) | " |
| 4 | 165 | 5 | 7.3e+1 | 1.9e+2 | 1.8e+1 | (0,4,6) | " |
| 5 | 165 | 5 | 1.2e+2 | 1.6e+2 | 1.9e+1 | (0,5,7) | " |
| 6 | 186 | 11 | 3.6e-1 | 4.3e-1 | 9.0e-2 | N/A | 500, 1000, 4000 |
| 7 | 186 | 8 | 3.6e-1 | 4.0e-1 | 9.1e-2 | (1,6,10) | " |
| 8 | 186 | 8 | 3.8e-1 | 3.7e-1 | 1.0e-1 | (5,7,10) | " |
| 9 | 186 | 6 | 3.9e-1 | 3.7e-1 | 9.7e-2 | (1,5,6,7,10) | " |
| 10 | 291 | 5 | 7.3e+0 | 3.8e+1 | 1.5e+0 | N/A | 200, 500, 1000 |
| 11 | 291 | 4 | 3.0e+1 | 2.0e+1 | 3.4e+0 | (0) | " |
| 12 | 291 | 3 | 2.2e+1 | 2.5e+1 | 4.5e+0 | (1,4) | " |
| 13 | 291 | 3 | 2.2e+1 | 2.5e+1 | 5.0e+0 | (2,4) | " |
| 14 | 291 | 2 | 1.7e+1 | 3.1e+1 | 4.6e+0 | (2,3,4) | " |
| 15 | 294 | 4 | 1.3e+1 | 2.8e+2 | 2.5e+0 | N/A | 500, 1000, 4000 |
| 16 | 294 | 3 | 3.4e+1 | 2.6e+2 | 6.9e+0 | (0) | " |
| 17 | 294 | 2 | 1.4e+2 | 1.5e+2 | 1.7e+1 | (0,1) | " |
| 18 | 294 | 2 | 4.5e+1 | 2.5e+2 | 6.9e+0 | (0,2) | " |
| 19 | 294 | 1 | 2.2e+2 | 7.3e+1 | 1.4e+1 | (0,1,2) | " |

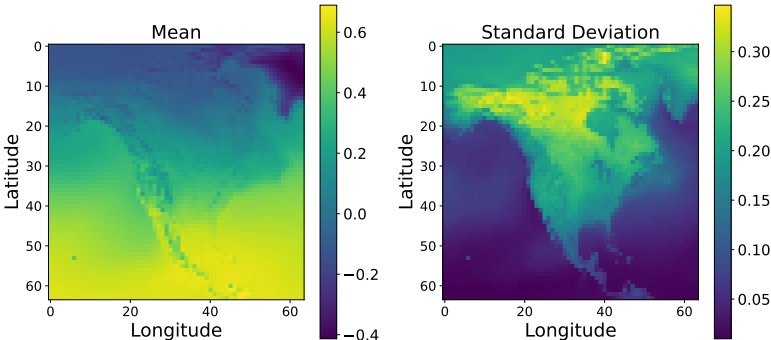

Figure 6: Normalized 2 m temperature distribution in January, 2024

## G   DESCRIPTION OF COMPOSITE STRUCTURE FAILURE DATASETS

We construct a dataset based on the **open-hole compression (OHC) test** of composite laminates, a standard experiment where a plate with a hole is compressed to study how stress concentrates and leads to material failure. The specimen is a Carbon Fibre Reinforced Plastic (CFRP) plate with in-plane dimensions $l_x = 50\,\text{mm}$, $l_y = 100\,\text{mm}$, and thickness $t_z$. A circular hole is introduced at relative coordinates $(r_x, r_y)$, where $r_x \in [0.3, 0.5]$, $r_y = 0.5$, and with radius $r_h \in [5, 12]\,\text{mm}$. The plate is clamped at the $y = 0$ face and subjected to compressive loading at the $y = l_y$ face.

The design variables include the relative x-coordinate of the hole center $r_x$ and the hole radius $r_h$. The output is the maximum failure index over the entire specimen. The failure index is a dimensionless measure of material safety: a value close to zero indicates safe loading, while a value exceeding one means the material has reached its limit and failure is expected to initiate.

Composite laminates consist of multiple plies stacked in the thickness direction. The laminate stacking sequence $\delta$ specifies the fiber orientation angles of all plies, and these orientations have a significant impact on how the laminate carries load and resists failure. In the early stages of industrial design for large composite structures, it is often impractical to fully account for the influence of $\delta$ on structural failure behavior due to its complexity and design cost. As a result, $\delta$ is treated as an unobservable variable, introducing intrinsic uncertainty into the failure prediction problem.

Overall, this dataset provides a controlled yet challenging benchmark for uncertainty quantification in composite failure prediction.

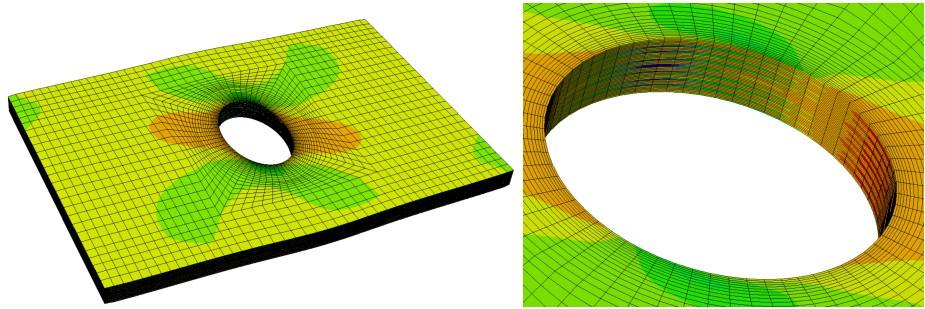

Figure 7: Failure index distribution in a composite specimen under an open-hole compression test. Left: the specimen is compressed uniaxially, leading to stress concentration around the hole. Right: a magnified view of the hole region, showing multiple layers with different fiber orientations. These orientations cause variations in stress and failure index values, with higher values indicating a greater likelihood of failure.

## H    RESULTS FOR UCI REGRESSION TASKS

Algorithm 1 is used to estimate the the average noise level $E_{\text{noise}}$, variance of the mean, $V_{\text{mean}}$, and the variance of the noise level, $V_{\text{noise}}$. Both the original UCI datasets and the modified ones are evaluated. Each dataset is evaluated under three different sample sizes ($N_s$), as listed in Table 5. For each case, 80% of the data is randomly selected for training, and the remaining 20% is used for testing.

For all the models, there is three hidden layers in the neural network, with 128 neurons in each layer; the dimension of features is 32. There is one hidden layers in the $g_\beta$ neural network of HVBLL, with 32 neurons in each layer. The initial learning rate is $0.01$, the learning rate gradually reduces during the training of 10,000 epochs. Adam optimizer is used for training. All the experiments are conducted on a NVIDIA A10 GPU. The performance of the models on the test sets in different cases is presented in Table 6 - 13.

Table 6: NLL results for UCI regression tasks (UCI ID = 165, $N_s = 200$)

| Dataset ID | 1 | 2 | 3 | 4 | 5 |
|---|---|---|---|---|---|
| HVBLL | **3.48 ± 0.22** | **4.01 ± 0.19** | **3.92 ± 0.16** | **3.88 ± 0.13** | **4.09 ± 0.14** |
| VBLL | 5.21 ± 1.12 | 6.10 ± 1.54 | 5.72 ± 1.58 | 5.52 ± 0.66 | 4.87 ± 0.35 |
| BLL | 9.32 ± 1.50 | 9.66 ± 1.62 | 9.63 ± 1.64 | 9.71 ± 1.62 | 9.63 ± 1.60 |
| Dropout | 4.72 ± 1.01 | 5.91 ± 1.25 | 7.19 ± 1.66 | 6.31 ± 0.58 | 10.22 ± 2.68 |
| PNN | 22.11 ± 9.46 | 35.79 ± 24.62 | 34.52 ± 26.16 | 39.03 ± 25.68 | 30.78 ± 17.39 |
| SWAG | 5.15 ± 0.27 | 5.05 ± 0.39 | 4.92 ± 0.29 | 5.16 ± 0.30 | 5.05 ± 0.32 |
| DVI | 5.22 ± 2.18 | 8.76 ± 3.35 | 8.40 ± 6.07 | 8.05 ± 5.03 | 11.00 ± 6.56 |
| MDN | 7.39 ± 2.58 | 8.72 ± 1.76 | 9.40 ± 3.44 | 8.08 ± 2.73 | 10.97 ± 3.25 |

Table 7: NLL results for UCI regression tasks (UCI ID = 165, $N_s = 1000$)

| Dataset ID | 1 | 2 | 3 | 4 | 5 |
|---|---|---|---|---|---|
| HVBLL | 3.59 ± 0.14 | **3.84 ± 0.12** | **3.89 ± 0.08** | **3.74 ± 0.09** | **4.05 ± 0.06** |
| VBLL | 3.89 ± 0.34 | 4.77 ± 0.42 | 4.27 ± 0.29 | 4.70 ± 0.34 | 4.12 ± 0.12 |
| BLL | 6.29 ± 0.38 | 7.43 ± 0.47 | 7.67 ± 0.48 | 7.91 ± 0.49 | 7.91 ± 0.46 |
| Dropout | 3.46 ± 0.48 | 5.09 ± 1.13 | 5.44 ± 0.59 | 4.77 ± 0.47 | 9.87 ± 0.74 |
| PNN | 9.23 ± 3.99 | 11.33 ± 12.08 | 6.81 ± 4.05 | 8.97 ± 4.99 | 12.40 ± 10.77 |
| SWAG | 4.82 ± 0.15 | 4.91 ± 0.23 | 4.98 ± 0.24 | 5.13 ± 0.28 | 4.96 ± 0.27 |
| DVI | 3.58 ± 0.98 | 4.94 ± 2.19 | 4.09 ± 0.34 | 4.13 ± 0.97 | 4.27 ± 0.50 |
| MDN | **3.20 ± 0.27** | 4.34 ± 1.18 | 3.98 ± 0.28 | 3.88 ± 0.30 | 4.46 ± 0.63 |

Table 8: NLL results for UCI regression tasks (UCI ID = 186, $N_s = 500$)

| Dataset ID | 1 | 2 | 3 | 4 |
|---|---|---|---|---|
| HVBLL | **1.14 ± 0.06** | **1.20 ± 0.09** | **1.31 ± 0.14** | **1.25 ± 0.09** |
| VBLL | 1.67 ± 0.19 | 1.67 ± 0.17 | 1.84 ± 0.20 | 2.01 ± 0.18 |
| BLL | 1.96 ± 0.01 | 1.97 ± 0.01 | 1.97 ± 0.01 | 1.97 ± 0.01 |
| Dropout | 4.51 ± 1.19 | 4.37 ± 1.05 | 4.54 ± 0.71 | 4.72 ± 1.37 |
| PNN | 18.98 ± 20.67 | 21.26 ± 32.79 | 14.20 ± 16.34 | 3.99 ± 2.69 |
| SWAG | 1.26 ± 0.19 | 1.48 ± 0.31 | 1.40 ± 0.27 | 1.60 ± 0.45 |
| DVI | 1.45 ± 0.20 | 1.68 ± 1.02 | 1.45 ± 0.17 | 2.55 ± 2.87 |
| MDN | 2.94 ± 0.85 | 4.21 ± 1.01 | 4.37 ± 0.92 | 3.62 ± 0.55 |

Table 9: NLL results for UCI regression tasks (UCI ID = 186, $N_s = 4000$)

| Dataset ID | 1 | 2 | 3 | 4 |
|---|---|---|---|---|
| HVBLL | **1.11 ± 0.04** | **1.12 ± 0.04** | **1.20 ± 0.03** | **1.21 ± 0.03** |
| VBLL | 1.17 ± 0.02 | 1.18 ± 0.02 | 1.22 ± 0.02 | 1.24 ± 0.01 |
| BLL | 1.96 ± 0.00 | 1.97 ± 0.00 | 1.96 ± 0.00 | 1.97 ± 0.00 |
| Dropout | 5.83 ± 0.65 | 7.30 ± 1.21 | 6.11 ± 0.82 | 8.90 ± 1.62 |
| PNN | 7.29 ± 4.59 | 4.14 ± 5.01 | 9.56 ± 12.41 | 7.64 ± 15.06 |
| SWAG | 1.25 ± 0.11 | 1.41 ± 0.11 | 1.34 ± 0.10 | 1.63 ± 0.21 |
| DVI | 1.16 ± 0.07 | 1.15 ± 0.06 | 1.22 ± 0.06 | 1.24 ± 0.07 |
| MDN | 1.43 ± 0.19 | 3.25 ± 0.41 | 3.11 ± 0.32 | 2.48 ± 0.15 |

Table 10: NLL results for UCI regression tasks (UCI ID = 291, $N_\mathrm{s} = 200$)

| Dataset ID | 1 | 2 | 3 | 4 | 5 |
|---|---|---|---|---|---|
| HVBLL | $\mathbf{2.79 \pm 0.20}$ | $\mathbf{3.21 \pm 0.13}$ | $\mathbf{3.11 \pm 0.17}$ | $\mathbf{3.16 \pm 0.17}$ | $\mathbf{3.16 \pm 0.13}$ |
| VBLL | $2.99 \pm 0.58$ | $3.39 \pm 0.28$ | $3.18 \pm 0.36$ | $3.42 \pm 0.51$ | $3.25 \pm 0.21$ |
| BLL | $13.54 \pm 1.72$ | $13.78 \pm 1.90$ | $12.13 \pm 1.52$ | $10.41 \pm 2.12$ | $8.09 \pm 1.51$ |
| Dropout | $2.94 \pm 0.06$ | $3.32 \pm 0.15$ | $3.16 \pm 0.09$ | $\mathbf{3.16 \pm 0.10}$ | $3.19 \pm 0.07$ |
| PNN | $6.41 \pm 1.87$ | $6.41 \pm 2.61$ | $5.66 \pm 2.38$ | $5.35 \pm 1.63$ | $4.15 \pm 0.68$ |
| SWAG | $3.68 \pm 1.07$ | $7.99 \pm 2.02$ | $7.59 \pm 3.09$ | $6.47 \pm 2.08$ | $9.48 \pm 2.64$ |
| DVI | $3.06 \pm 0.13$ | $3.41 \pm 0.15$ | $3.26 \pm 0.12$ | $3.30 \pm 0.12$ | $3.30 \pm 0.09$ |
| MDN | $5.96 \pm 0.10$ | $5.92 \pm 0.00$ | $5.92 \pm 0.00$ | $5.92 \pm 0.00$ | $5.86 \pm 0.33$ |

Table 11: NLL results for UCI regression tasks (UCI ID = 291, $N_\mathrm{s} = 1000$)

| Dataset ID | 1 | 2 | 3 | 4 | 5 |
|---|---|---|---|---|---|
| HVBLL | $2.69 \pm 0.19$ | $\mathbf{3.23 \pm 0.04}$ | $\mathbf{3.11 \pm 0.05}$ | $3.17 \pm 0.07$ | $3.20 \pm 0.07$ |
| VBLL | $\mathbf{2.46 \pm 0.13}$ | $3.40 \pm 0.15$ | $3.20 \pm 0.15$ | $3.34 \pm 0.18$ | $3.30 \pm 0.09$ |
| BLL | $7.33 \pm 0.37$ | $7.66 \pm 0.39$ | $7.11 \pm 0.48$ | $6.53 \pm 0.42$ | $5.66 \pm 0.36$ |
| Dropout | $2.89 \pm 0.05$ | $3.25 \pm 0.03$ | $3.12 \pm 0.03$ | $\mathbf{3.13 \pm 0.04}$ | $\mathbf{3.18 \pm 0.05}$ |
| PNN | $3.36 \pm 0.29$ | $3.77 \pm 0.17$ | $3.60 \pm 0.37$ | $3.60 \pm 0.23$ | $3.39 \pm 0.15$ |
| SWAG | $2.57 \pm 0.08$ | $9.96 \pm 2.29$ | $5.66 \pm 0.85$ | $6.34 \pm 1.21$ | $8.18 \pm 1.73$ |
| DVI | $2.90 \pm 0.11$ | $3.28 \pm 0.06$ | $3.16 \pm 0.06$ | $3.23 \pm 0.06$ | $3.27 \pm 0.06$ |
| MDN | $5.92 \pm 0.00$ | $5.92 \pm 0.00$ | $5.92 \pm 0.00$ | $5.92 \pm 0.00$ | $5.52 \pm 0.88$ |

Table 12: NLL results for UCI regression tasks (UCI ID = 294, $N_\mathrm{s} = 500$)

| Dataset ID | 1 | 2 | 3 | 4 | 5 |
|---|---|---|---|---|---|
| HVBLL | $\mathbf{2.91 \pm 0.10}$ | $3.46 \pm 0.03$ | $\mathbf{4.03 \pm 0.06}$ | $3.51 \pm 0.05$ | $\mathbf{4.17 \pm 0.06}$ |
| VBLL | $2.94 \pm 0.07$ | $\mathbf{3.44 \pm 0.03}$ | $\mathbf{4.03 \pm 0.06}$ | $\mathbf{3.49 \pm 0.05}$ | $\mathbf{4.17 \pm 0.06}$ |
| BLL | $33.78 \pm 6.26$ | $27.19 \pm 4.94$ | $31.46 \pm 4.93$ | $23.15 \pm 4.42$ | $29.65 \pm 3.77$ |
| Dropout | $4.17 \pm 0.06$ | $4.21 \pm 0.09$ | $4.29 \pm 0.04$ | $4.23 \pm 0.05$ | $4.37 \pm 0.04$ |
| PNN | $3.46 \pm 0.45$ | $3.92 \pm 0.21$ | $5.14 \pm 1.75$ | $3.80 \pm 0.15$ | $4.60 \pm 0.51$ |
| SWAG | $6.23 \pm 0.71$ | $5.70 \pm 0.47$ | $6.08 \pm 0.82$ | $5.97 \pm 0.84$ | $6.03 \pm 0.70$ |
| DVI | $3.90 \pm 0.39$ | $3.94 \pm 0.34$ | $4.14 \pm 0.12$ | $3.69 \pm 0.16$ | $4.29 \pm 0.14$ |
| MDN | $5.94 \pm 0.02$ | $5.96 \pm 0.02$ | $5.93 \pm 0.01$ | $6.13 \pm 0.55$ | $5.93 \pm 0.01$ |

Table 13: NLL results for UCI regression tasks (UCI ID = 294, $N_\mathrm{s} = 4000$)

| Dataset ID | 1 | 2 | 3 | 4 | 5 |
|---|---|---|---|---|---|
| HVBLL | $\mathbf{2.91 \pm 0.06}$ | $3.46 \pm 0.05$ | $4.03 \pm 0.01$ | $3.54 \pm 0.03$ | $\mathbf{4.18 \pm 0.02}$ |
| VBLL | $2.93 \pm 0.05$ | $3.43 \pm 0.05$ | $4.03 \pm 0.02$ | $3.51 \pm 0.04$ | $\mathbf{4.18 \pm 0.02}$ |
| BLL | $11.90 \pm 0.81$ | $14.78 \pm 0.73$ | $20.96 \pm 0.90$ | $13.13 \pm 0.46$ | $22.26 \pm 0.60$ |
| Dropout | $4.19 \pm 0.05$ | $4.18 \pm 0.04$ | $4.30 \pm 0.03$ | $4.19 \pm 0.05$ | $4.35 \pm 0.03$ |
| PNN | $3.24 \pm 0.18$ | $3.63 \pm 0.07$ | $4.30 \pm 0.04$ | $3.73 \pm 0.09$ | $4.43 \pm 0.08$ |
| SWAG | $5.26 \pm 0.50$ | $5.00 \pm 0.58$ | $5.09 \pm 0.43$ | $6.54 \pm 2.93$ | $5.07 \pm 0.33$ |
| DVI | $2.97 \pm 0.11$ | $\mathbf{3.42 \pm 0.08}$ | $\mathbf{4.00 \pm 0.02}$ | $\mathbf{3.47 \pm 0.03}$ | $\mathbf{4.18 \pm 0.02}$ |
| MDN | $5.94 \pm 0.02$ | $5.97 \pm 0.13$ | $5.93 \pm 0.01$ | $5.92 \pm 0.01$ | $5.96 \pm 0.04$ |

## I RESULTS FOR ERA5 REGRESSION TASKS

Algorithm 1 is used to estimate the the average noise level $E_{\text{noise}}$, variance of the mean, $V_{\text{mean}}$, and the variance of the noise level, $V_{\text{noise}}$. Each dataset is evaluated under three different sample sizes ($N_{\text{s}}$), as shown in the label of each column. For each case, 80% of the data is randomly selected for training, and the remaining 20% is used for testing.

For all the models, there is three hidden layers in the neural network, with 128 neurons in each layer; the dimension of features is 64. There is one hidden layers in the $g_\beta$ neural network of HVBLL, with 32 neurons in each layer. The initial learning rate is 0.01, the learning rate gradually reduces during the training of 10,000 epochs. Adam optimizer is used for training. All the experiments are conducted on a NVIDIA A10 GPU. The performance of the models on the test sets in different cases is presented in Table 14 - 16.

Table 14: NLL results for ERA5 regression tasks

| Model | 500 | 4000 | 20000 |
|---|---|---|---|
| HVBLL | $\mathbf{-0.10 \pm 0.16}$ | $\mathbf{-0.84 \pm 0.12}$ | $-0.68 \pm 0.31$ |
| VBLL | $0.06 \pm 0.07$ | $-0.21 \pm 0.00$ | $-0.18 \pm 0.06$ |
| BLL | $2.42 \pm 0.00$ | $2.42 \pm 0.00$ | $2.42 \pm 0.00$ |
| Dropout | $2.44 \pm 0.55$ | $0.83 \pm 1.01$ | $1.47 \pm 2.80$ |
| PNN | $26.55 \pm 13.88$ | $7.38 \pm 7.61$ | $-0.65 \pm 0.05$ |
| SWAG | $0.84 \pm 0.41$ | $-0.17 \pm 0.08$ | $-0.04 \pm 0.06$ |
| DVI | $0.35 \pm 0.04$ | $0.32 \pm 0.02$ | $0.32 \pm 0.01$ |
| MDN | $2.95 \pm 1.48$ | $-0.83 \pm 0.18$ | $\mathbf{-1.32 \pm 0.07}$ |

Table 15: MAE results for ERA5 regression tasks

| Model | 500 | 4000 | 20000 |
|---|---|---|---|
| HVBLL | $\mathbf{0.13 \pm 0.01}$ | $\mathbf{0.07 \pm 0.01}$ | $0.08 \pm 0.03$ |
| VBLL | $0.16 \pm 0.02$ | $\mathbf{0.07 \pm 0.00}$ | $0.07 \pm 0.02$ |
| BLL | $0.18 \pm 0.01$ | $0.14 \pm 0.03$ | $0.15 \pm 0.04$ |
| Dropout | $\mathbf{0.13 \pm 0.01}$ | $0.09 \pm 0.01$ | $0.08 \pm 0.01$ |
| PNN | $\mathbf{0.13 \pm 0.01}$ | $0.11 \pm 0.01$ | $0.10 \pm 0.00$ |
| SWAG | $0.16 \pm 0.01$ | $0.17 \pm 0.02$ | $0.12 \pm 0.01$ |
| DVI | $0.28 \pm 0.01$ | $0.28 \pm 0.01$ | $0.28 \pm 0.00$ |
| MDN | $0.14 \pm 0.01$ | $\mathbf{0.07 \pm 0.00}$ | $\mathbf{0.06 \pm 0.00}$ |

## J RESULTS FOR COMPOSITE STRUCTURE FAILURE REGRESSION TASKS

Algorithm 1 is used to estimate the the average noise level $E_{\text{noise}}$, variance of the mean, $V_{\text{mean}}$, and the variance of the noise level, $V_{\text{noise}}$. Each dataset is evaluated under three different sample sizes ($N_{\text{s}}$), as shown in the label of each column. For each case, 80% of the data is randomly selected for training, and the remaining 20% is used for testing.

For all the models, there is three hidden layers in the neural network, with 128 neurons in each layer; the dimension of features is 64. There is one hidden layers in the $g_\beta$ neural network of HVBLL, with 32 neurons

Table 16: CRPS results for ERA5 regression tasks

| Model | 500 | 4000 | 20000 |
|---|---|---|---|
| HVBLL | $\mathbf{0.10 \pm 0.01}$ | $\mathbf{0.05 \pm 0.01}$ | $0.07 \pm 0.02$ |
| VBLL | $0.13 \pm 0.01$ | $0.08 \pm 0.00$ | $0.09 \pm 0.01$ |
| BLL | $1.05 \pm 0.00$ | $1.05 \pm 0.00$ | $1.05 \pm 0.00$ |
| Dropout | $\mathbf{0.10 \pm 0.01}$ | $0.07 \pm 0.01$ | $\mathbf{0.06 \pm 0.01}$ |
| PNN | $\mathbf{0.10 \pm 0.01}$ | $0.08 \pm 0.00$ | $0.07 \pm 0.00$ |
| SWAG | $0.12 \pm 0.01$ | $0.12 \pm 0.01$ | $0.11 \pm 0.01$ |
| DVI | $0.19 \pm 0.01$ | $0.19 \pm 0.00$ | $0.19 \pm 0.00$ |
| MDN | $\mathbf{0.11 \pm 0.01}$ | $\mathbf{0.05 \pm 0.00}$ | $\mathbf{0.05 \pm 0.00}$ |

in each layer. The initial learning rate is $0.01$, the learning rate gradually reduces during the training of 10,000 epochs. Adam optimizer is used for training. All the experiments are conducted on a NVIDIA A10 GPU. The performance of the models on the test sets in different cases is presented in Table 17 - 19.

Table 17: NLL results for laminate regression tasks

| Model | 500 | 1000 | 4000 |
|---|---|---|---|
| HVBLL | $\mathbf{-0.73 \pm 0.06}$ | $\mathbf{-0.74 \pm 0.04}$ | $\mathbf{-0.75 \pm 0.02}$ |
| VBLL | $-0.16 \pm 0.02$ | $-0.15 \pm 0.00$ | $-0.15 \pm 0.00$ |
| BLL | $2.42 \pm 0.00$ | $2.42 \pm 0.00$ | $2.42 \pm 0.00$ |
| Dropout | $2.42 \pm 0.77$ | $4.13 \pm 0.51$ | $17.50 \pm 2.97$ |
| PNN | $0.47 \pm 0.92$ | $0.15 \pm 0.48$ | $-0.45 \pm 0.05$ |
| SWAG | $0.40 \pm 0.65$ | $0.30 \pm 0.43$ | $0.24 \pm 0.31$ |
| DVI | $-0.61 \pm 0.06$ | $-0.59 \pm 0.03$ | $-0.62 \pm 0.02$ |
| MDN | $-0.59 \pm 0.09$ | $-0.62 \pm 0.09$ | $-0.63 \pm 0.10$ |

Table 18: MAE results for laminate regression tasks

| Model | 500 | 1000 | 4000 |
|---|---|---|---|
| HVBLL | $\mathbf{0.10 \pm 0.01}$ | $\mathbf{0.10 \pm 0.00}$ | $\mathbf{0.10 \pm 0.00}$ |
| VBLL | $0.11 \pm 0.01$ | $\mathbf{0.10 \pm 0.00}$ | $\mathbf{0.10 \pm 0.00}$ |
| BLL | $0.11 \pm 0.01$ | $0.11 \pm 0.00$ | $\mathbf{0.10 \pm 0.00}$ |
| Dropout | $\mathbf{0.10 \pm 0.00}$ | $\mathbf{0.10 \pm 0.00}$ | $\mathbf{0.10 \pm 0.00}$ |
| PNN | $0.11 \pm 0.01$ | $0.11 \pm 0.00$ | $\mathbf{0.10 \pm 0.00}$ |
| SWAG | $\mathbf{0.10 \pm 0.00}$ | $\mathbf{0.10 \pm 0.00}$ | $\mathbf{0.10 \pm 0.00}$ |
| DVI | $0.11 \pm 0.01$ | $0.11 \pm 0.00$ | $0.11 \pm 0.00$ |
| MDN | $0.11 \pm 0.00$ | $\mathbf{0.10 \pm 0.00}$ | $\mathbf{0.10 \pm 0.00}$ |

Table 19: CRPS results for laminate regression tasks

| Model | 500 | 1000 | 4000 |
|---|---|---|---|
| HVBLL | $\mathbf{0.07 \pm 0.00}$ | $\mathbf{0.07 \pm 0.00}$ | $\mathbf{0.07 \pm 0.00}$ |
| VBLL | $0.09 \pm 0.00$ | $0.09 \pm 0.00$ | $0.09 \pm 0.00$ |
| BLL | $1.05 \pm 0.00$ | $1.05 \pm 0.00$ | $1.04 \pm 0.00$ |
| Dropout | $\mathbf{0.08 \pm 0.00}$ | $\mathbf{0.08 \pm 0.00}$ | $0.09 \pm 0.00$ |
| PNN | $\mathbf{0.08 \pm 0.01}$ | $\mathbf{0.08 \pm 0.00}$ | $\mathbf{0.07 \pm 0.00}$ |
| SWAG | $\mathbf{0.08 \pm 0.01}$ | $\mathbf{0.08 \pm 0.00}$ | $\mathbf{0.08 \pm 0.00}$ |
| DVI | $\mathbf{0.08 \pm 0.00}$ | $\mathbf{0.08 \pm 0.00}$ | $\mathbf{0.07 \pm 0.00}$ |
| MDN | $\mathbf{0.07 \pm 0.00}$ | $\mathbf{0.07 \pm 0.00}$ | $\mathbf{0.07 \pm 0.00}$ |