# OpenReview forum: "Heteroscedastic Variational Bayesian Last Layers: Modeling Input-Dependent Noise in Sparse-Data Regression"
_ICLR.cc/2026/Conference — ICLR 2026 Conference Withdrawn Submission_

### Official Review · Reviewer_3Jrp · 2025-10-30

**Soundness:** 2
**Presentation:** 1
**Contribution:** 2
**Rating:** 2
**Confidence:** 4

**Summary:**

Aiming to address the drawbacks of Variational Bayesian Last Layer (VBLL), i.e., assuming existence of homoscedastic noise and sufficient data, the authors propose to model heteroscedastic noise within VBLL and suggest a clustering-based initialization for the prior noise variance for robust performance. Experimental results on toy datasets and several real-world datasets on UCI and ERA5 demonstrate the benefits of the proposed idea when compared with six baselines.

**Strengths:**

- The proposed idea seems to address a relevant problem of modeling input-dependent/heteroscedastic aleatoric uncertainty within the promising approach for efficient uncertainty estimation - VBLL.

- Technically, the authors recognize the issue of misspecified prior noise when modeling the heteroscedastic noise and suggest a clustering-based initilization to mitigate this issue.

- Ablation study on the effects of misspecified noise initilization is insightful and convincing.

**Weaknesses:**

Presentation:
- The presentation flow of the draft is a bit hard follow. Especially in the prelinaries section, it seems the motivation and connection between each sub-sections are missing.

- It would be also more readable if the part about prior works can be separated from the introduction and used to position this work in the leterature more clearly.

Method:
- the proposed extension of modeling heteroscedastic noise in VBLL is technically incremental and the cases for classification and generative classification have not been discussed and considered.

- The clustering-based approach requires an additional dataset for the initilization. Does it add more hyper-parameters or complexity for this approch?

- There is a lack of discussing the roles of aleatoric and epistemic uncertainty based on the given formulation.

- It seems that VBLL can be connected to Gaussian Processes (GPs) in view of the function space [1], it would be inspring for general readers to add a discussion about the commonality and difference between them.

Experiments:
- Missing a relevant baseline: Gaussian Processes via Neural Tangent Kernel for the last layer, in which [1] has implemented for a robot inverse dynamics regression task.

- It would be more clear to add results when using all the data points in the real-world regression tasks.

- It would be more convincing to add the variance to the result table from different runs/sample sizes.

[1] Lee, J., Feng, J., Humt, M., Müller, M. G., & Triebel, R. (2022, January). Trust your robots! predictive uncertainty estimation of neural networks with sparse gaussian processes. In Conference on Robot Learning (pp. 1168-1179). PMLR.

**Questions:**

see above

---

### Official Review · Reviewer_iYmP · 2025-10-31

**Soundness:** 3
**Presentation:** 2
**Contribution:** 2
**Rating:** 4
**Confidence:** 3

**Summary:**

This paper proposes a new extension of the Variational Bayesian Last Layer (VBLL) framework called Heteroscedastic VBLL (HVBLL)  that can explicitly model heteroscedastic (input-dependent) noise while retaining VBLL’s computational efficiency and sampling-free properties. The main idea of the proposed HVBLL is to replace the constant-homoscedastic noise assumption in VBLL with an input-dependent Gaussian distribution, with the variance $\sigma(x)^2$ parameterized by a neural network. Moreover, the authors demonstrate the sensitivity of both VBLL and their proposed HVBLL to noise priors in sparse-data scenarios, and design a clustering-based noise-level estimation algorithm to infer a more reasonable noise prior. Empirically, the proposed method shows strong performance (e.g., captures heteroscedastic noise, models noise prior) across both synthetic and real-world datasets (UCI, ERA5, and a composite structure failure dataset). The HVBLL consistently outperforms baselines such as VBLL, MC-Dropout, SWAG, BLL, DVI, and PNN in both accuracy and uncertainty metrics (NLL, MAE, CRPS).

**Strengths:**

1. The proposed HVBLL generalizes VBLL to heteroscedastic noise settings with a flexible parametrization for the input-dependent noise variance (modeling as a neural network) while retaining deterministic and sampling-free training.
2. The proposed clustering-based noise prior estimation is simple ($Mean(v_i)$) yet powerful and directly addresses a key sensitivity issue in sparse-data BNNs.
3. Empirically, experimental evaluation involving multiple tasks (heteroscedastic noise capture, noise priors, benchmark uci regression) to illustrate the effectiveness of the proposed method. HVBLL achieves consistently lower NLL and CRPS than VBLL and other baselines, especially in heteroscedastic and sparse-data cases (Tables 1–2 and 14–19).

**Weaknesses:**

1. While parameterizing heteroscedasticity with neural networks $g_{\beta}$ offers flexibility, it increases computational burden. The paper does not analyze the computational efficiency of the proposed method. Moreover, there is no ablation study for this $g_{\beta}$ to demonstrate its effect on the performance of HVBLL, which is the key difference between the proposed HVBLL and Vallina VBLL.
2. For uncertainty estimation, the paper reports NLL, MAE, and CRPS but does not provide calibration metrics such as ECE, which is an important uncertainty calibration metric.
3. Although the paper includes real-world datasets (UCI), it is unclear whether the method could scale to large real-world inputs (e.g., image classification tasks for CIFAR10/100, etc).
4. While Algorithm 1 performs well empirically, there is no theoretical analysis of its convergence or bias relative to ground-truth variance estimation. Moreover, the choice of cluster size directly affects the estimation of $E_{noise}$, however, the paper provides no sensitivity or ablation study for this parameter.
5. There are some typos that require further calibration, e.g., in eq.(9), the variance $g_{\beta}$ should be $\exp g_{\beta}$; In Fig 3, the equation reference in the figure is incorrect (eq.24-27) and should be eq.28-31.

**Questions:**

1. I believe the main distinction between your proposed HVBLL loss function and the VBLL loss function lies in modeling noise variance. $q(\epsilon) \sim N(0, \sigma^2)$ in Vallina VBLL and $q_{\beta}(\epsilon)\sim N(0, g_{\beta}(x))$ in your method. In your loss function (10), there is a KL term between the noise prior and approximation posterior; however, Eq. (4) for Vallina VBLL lacks such a KL term, containing only $\log p(\sigma^2)$. Why is this KL term omitted? What does $p(\sigma^2)$ represent?
2. In Eq.(5), such $\mathcal L(\theta, \eta, \sigma^2)$ is demonstrated as the ELBO; however, I believe this term is just the expected log-likelihood (because there is no KL term in this $\mathcal L$). What's the difference between these two?
3. Can you provide a theoretical justification or bias analysis of Algorithm 1’s estimation of $Mean(v_{i})$ relative to the true $E_{noise}$? I think this approximation accuracy should be highly correlated with the number of clusters. Can you provide a sensitivity analysis for this hyperparameter? Due to time constraints, demonstrating it on a toy experiment would suffice.
4. How does HVBLL scale in complexity and stability to higher-dimensional image classification tasks (e.g., CIFAR10/100, ≥ $10^3$ features)? All networks in your experiments are MLPs. Can HVBLL be efficiently integrated into other network architectures, such as CNNs?
5. You just used one-hidden-layer neural network for $g_{\beta}$, with 32 hidden units for real-world datasets. Is this simple architecture sufficient for high-dimensional inputs? How significantly does it impact the final performance of HVBLL? Could you provide the ablation study results?
6. What's the computational complexity of the HVBLL? How much computational overhead does this parameterized neural network $g_{\beta}$ introduce? Can you provide runtime comparisons with other methods?
7. It's unclear how AI and WR are calculated in Table 1-2. Can you provide the specific computational definitions?

---

### Official Review · Reviewer_Qh17 · 2025-11-02

**Soundness:** 2
**Presentation:** 2
**Contribution:** 2
**Rating:** 2
**Confidence:** 4

**Summary:**

The paper proposes a method based on the idea of Variational Bayesian Last Layer (VBLL) to estimate heteroschedastic noise in regression type problems within a sparse data regime. The paper proposes a clsuteting-based noise level estimation in this framework and demonstrates performance of the method on several synthetic and real world data sets. In terms of the methods the comparison is done with respect to Monte Carlo drop out. One of the main conclusions is that the prior on the noise has a significant effect in terms of the quality of uncertainty quantification.

**Strengths:**

Generally the problem is well posed and there is lack of computationally efficient methods for uncertainty quantification within neural networks context, especially when it comes to the estimation of heteroskedastic noise.

**Weaknesses:**

- The set up of the paper is not convincing. In particular, the proposed method outperforms other methods (such as MC dropout) in small data regimes. But I am not sure why neural networks in this regime would be chosen for modeling at all. As introduction states, Gaussian processes provide well calibrated uncertainty estimates, although they are computationally expensive. It seems like that would be a much more desirable modeling framework for such kinds of applications.

- The results suggest the there is a strong effect for the choice of the noise of the prior level on performance of the proposed method. Clustering-based noise estimation to set the prior is proposed, however, that does not seem to be rigorously analyzed. Essentially this is not sufficiently developed method to be used in diverse scenarios.

**Questions:**

- The literature review for the sparse data regimes can be stronger. In particular it would make sense then to compare to Gaussian processes. It is not clear from the paper why choose neural network setup for such cases as it does not seem to be optimal framework. This could be motivated perhaps by some critical applications which require fast real time inference, but nothing like that is considered in the experiments.

- How well does the method perform compared to more conventional methods for sparse data regimes?

- What are the failure modes of the proposed method to set prior hyperparamters?

Overall, the performance of the method seems good in the cases where neural networks would not be the first choice for modeling. Since the performance of the method depends so strongly on the choice of the hyperparamerers, this part of the approach needs to be developed further to account for the diverse scenarios, including more complex data sets.

---

### Note · Authors · 2025-12-02

I have read and agree with the venue's withdrawal policy on behalf of myself and my co-authors.